# Rationalizing the Dependence of Poly (Vinylidene Difluoride) (PVDF) Rheological Performance on the Nano-Silica

**DOI:** 10.3390/nano13061096

**Published:** 2023-03-18

**Authors:** Yi Cui, Yang Sui, Peng Wei, Yinan Lv, Chuanbo Cong, Xiaoyu Meng, Hai-Mu Ye, Qiong Zhou

**Affiliations:** 1Department of Materials Science and Engineering, New Energy and Material College, China University of Petroleum-Beijing, Beijing 102249, China; 2Beijing Key Laboratory of Failure, Corrosion, and Protection of Oil/Gas Facilities, China University of Petroleum-Beijing, Beijing 102249, China; 3State Key Laboratory of Tribology, Department of Mechanical Engineering, Tsinghua University, Beijing 100084, China

**Keywords:** poly (vinylidene difluoride), rheological behaviors, entanglements

## Abstract

Research on the rheological performance and mechanism of polymer nanocomposites (PNCs), mainly focuses on non-polar polymer matrices, but rarely on strongly polar ones. To fill this gap, this paper explores the influence of nanofillers on the rheological properties of poly (vinylidene difluoride) (PVDF). The effects of particle diameter and content on the microstructure, rheology, crystallization, and mechanical properties of PVDF/SiO_2_ were analyzed, by TEM, DLS, DMA, and DSC. The results show that nanoparticles can greatly reduce the entanglement degree and viscosity of PVDF (up to 76%), without affecting the hydrogen bonds of the matrix, which can be explained by selective adsorption theory. Moreover, uniformly dispersed nanoparticles can promote the crystallization and mechanical properties of PVDF. In summary, the viscosity regulation mechanism of nanoparticles for non-polar polymers, is also applicable to PVDF, with strong polarity, which is of great value for exploring the rheological behavior of PNCs and guiding the process of polymers.

## 1. Introduction

Most polymers are processed in a flowing state, so rheology is the most important property of the polymer process. Polymer nanocomposites (PNCs) have received wide attention from researchers in recent years, because they can combine the performance of polymers with the special properties of nanofillers [1,2]. Nanofillers, even at low volume fractions, can significantly improve the fluidity [3,4], crystallization, mechanical properties [5,6], thermal stability [7], barrier properties [8,9], optics [10,11], electrical [12], and other properties of the polymer matrix.

Studies [13] indicate that dispersing a small amount of nanofiller into the polymer matrix can cause a decrease in the polymer viscosity, to varying degrees. This phenomenon has been demonstrated in numerous PNCs [14,15,16,17,18], such as polystyrene (PS)/PS nanoparticles [17], polypropylene/nano-silica [19], and ultra-high molecular weight polyethylene/carbon nanotubes (CNTs) [15]. The explanation of this phenomenon mainly includes two aspects, namely constraint release theory (monodisperse system) and selective adsorption theory (polydisperse system). Mackay et al. [17,20,21,22,23] found that the nanofiller can reach a thermodynamically stable uniform dispersion state through reasonable dispersion methods, when the particle diameter and spacing are less than the polymer radius of gyration (Rg). At this time, the rapid diffusion of nanoparticles in the matrix will reduce the restraint of the molecular chain, so that the entanglement degree will decline. As a result, the viscosity of the polymer matrix will decrease macroscopically, which is called the phenomenon of “constraint release” [22]. The constraint release theory is only applicable to monodisperse systems, but for polydisperse systems, researchers have proposed a completely different theory, called selective adsorption. Jain [19] and Rastogi [15] believed that the surfaces of nanoparticles selectively adsorb polymer chains of relatively high molecular weight, resulting in a decrease in the molecular weight of the remaining polymer, thereby reducing entanglement and viscosity. Moreover, they extend this phenomenon to non-spherical CNT systems. In addition, He et al. [24] also theoretically demonstrated that polymers attached to the surface of nanoparticles have a higher molecular weight, through molecular dynamics simulations.

Unfortunately, the current research on the viscosity reduction in PNCs by nanoparticles, mainly focuses on non-polar matrix systems. Meanwhile, polar polymers account for a larger proportion of practical applications than non-polar polymers, so it is essential to investigate the effects of nanoparticles on the rheological behavior of strongly polar systems. However, it is difficult to study the viscosity reduction in PNCs in polymer matrices with strong intermolecular forces, because the viscosity is greatly affected by polarity and hydrogen bonding [25,26,27,28].

Poly (vinylidene difluoride) (PVDF) is a kind of fluororesin, with broad development prospects [29,30,31]. It has many intramolecular (formed by H and F atoms in the same macromolecule) and intermolecular hydrogen bonds (between H and F atoms from different macromolecules) [32], and strong polarity, and its main chain has a certain degree of flexibility, which make it a typical representative of polar polymers [32]. Haddadi et al. [25] and Tang et al. [33], respectively, investigated the influences of nano-silica and multi-wall carbon nanotubes (MWCNTs), on the rheological behavior of PVDF. The results indicated that the viscosity of PVDF increased with the increase in nano-silica and MWCNTs. However, their study only utilized nanofillers of specific size and content, so systematic studies on the rheological behavior of PVDF-based nanocomposites are still lacking. Therefore, in this paper, we utilized PVDF as the matrix and SiO_2_ nanoparticles of three different diameters and eleven different contents, as nanofillers, to fabricate uniformly dispersed PNC systems, by the rapid precipitation method. The effects of particle diameter and content of nanoparticles on the rheological behavior of the PNCs were investigated, and the reasons for the viscosity changes were analyzed by FTIR and DMA, from the perspective of hydrogen bonds and entanglement densities, respectively. In addition, this paper reveals the influence of nanoparticles on the crystallization behavior and mechanical properties of PVDF.

## 2. Experimental Sections

### 2.1. Raw Materials

The brand of poly (vinylidene difluoride) (PVDF) used in this study, was 6020 (Solef Company, Greenville, SC, USA, Rg ≈ 16.9 nm, ρ ≈ 1.78 g/cm^3^), with a molecular weight of 700 kDa and PDI of 2.1~2.4. There is the β-phase in our PVDF, which can be confirmed from the XRD pattern (Appendix A) [34], and thus, PVDF in this study is typically polar. 2-Propenoic acid, 2-methyl-, 3-(trimethoxysilyl) propyl ester (KH570), pentaerythritol tetrakys 3-(3,5-ditert-butyl-4-hydroxyphenyl) propionate ethanol (antioxidants 1010), N, N-dimethylformamide (DMF), and nano-silicas (ρ ≈ 0.11 g/cm^3^) with different diameters (20 nm, 30 nm, and 50 nm), were all purchased from Aladdin (Shanghai, China).

### 2.2. Modification of Silica Nanoparticles

Nano-silica was surface modified by KH570, and the mechanism is shown as Figure 1 [35]. The chemical formula is CH_2_=C(CH_3_)COO(CH_2_)Si(OCH_3_)_3_ and its reaction with nano-SiO_2_ consists of three steps: (1) the –R group attached to the silicon atom hydrolyzes, to form Si–OH oligosiloxane; (2) the Si–OH in oligosiloxane forms a hydrogen bond with the –OH on the surface of nano-SiO_2_; (3) a covalent bond with SiO_2_ is formed, along with the dehydration reaction during the heating process. 

The nano-silicas with different particle diameters were dried in an oven at 90 °C, for 16 h. One gram of SiO_2_ nanoparticles was dispersed in 100 mL ethanol and then sonicated for 1 h. After that, 50 mg KH570 was added into the ethanol dispersion dropwise. Subsequently, the mixed liquid was heated and stirred in an oil bath at 70 °C, for 6 h. The suspension was first centrifuged at 8000 r/min, to obtain the supernatant; then the supernatant was centrifuged at 15,000 r/min, to obtain the precipitations. Finally, the precipitate was placed in an oven at 70 °C, for 24 h, to obtain modified SiO_2_ particles.

### 2.3. Preparation of PVDF/SiO_2_ Dispersion and Films

An appropriate amount of modified nano-silica was weighed and dispersed in 100 mL DMF by ultrasonication, and then 10 g PVDF powder and 30 mg antioxidant 1010 were added to the dispersion, respectively. The mixed solution was placed in a water bath at 40 °C, and heated with stirring for 18 h, to obtain a uniform PVDF/SiO_2_ dispersion. The mass fraction of nano-SiO_2_ increased from 0.1 wt% to 2.0 wt%, while its volume fraction increased from 1.59 vol% to 24.25 vol% (the conversion between mass and volume fraction is provided in Appendix A). PVDF/SiO_2_ composite films were prepared by a method of rapid precipitation of the PVDF/SiO_2_ dispersion. Specifically, the dispersion was added dropwise to 1000 mL of distilled water with rapid stirring, to achieve the precipitate of the PVDF/SiO_2_ composite material. The filtered precipitate was shredded and placed in a vacuum oven at 70 °C, for 48 h, to remove the residual solvent. The dried samples were pressed at a pressure of 90 MPa, for 10 min, at 190 °C, using an automatic pressure precision machine (20T, Guangdong Lina Company, China). Subsequently, they were pressed at 20 °C, with the same pressure and time, to obtain the desired film samples for a series of characterizations.

### 2.4. Characterization of PVDF/SiO_2_ Composites

#### 2.4.1. Viscosity of PVDF/SiO_2_ Solutions

The prepared solution was placed in a cylindrical sample vial, with a size of 27 mm × 72.5 mm, and its shear viscosity at different shear rates was measured at 40 °C, using a rotational viscometer (PREL40003, Fungilab, Barcelona, Spain). The shear rate varied from 0.034 s^−1^ to 17 s^−1^, while the measured solution viscosity was always within 15% to 95% of the measurement range, to keep the measurement result accurate.

#### 2.4.2. Transmission Electron Microscopy (TEM)

TEM (F20, Tokyo, Japan) was utilized to characterize the diameter of the modified silica particles and their dispersion state in the PVDF matrix. The dispersion of nanoparticles in ethanol, with a concentration of 0.1 mg/mL, was dropped onto ultra-thin carbon film, and was dried in a dust-free oven at 70 °C for 1 h, to obtain the tested sample of pure SiO_2_ nanoparticles. The tested sample of the PVDF/SiO_2_ composite, with a thickness of 80 nm, was fabricated through an ultramicrotome (Leica EM UC7, Weztlar, Germany), and then placed on an ultrathin carbon film.

#### 2.4.3. Light Scattering

The distribution of nanoparticle diameter was revealed by dynamic light scattering (DLS) (Anton-Paar, Litesizer 500, Vienna, Austria). A volume of 3 mL of the SiO_2_ particle dispersion, with a concentration of 0.02 mg/mL, was placed in a quartz cuvette, and 60 rounds of DLS tests were performed, at 25 °C, with 10 s per round. The refractive index of the nanoparticles was 1.4900, the absorption coefficient was 0.0020, the refractive index of ethanol was 1.3577, and the viscosity was 1.0393 mPa·s.

The light intensity and scattering vector of PVDF/SiO_2_ dispersion (10 wt%), were measured at different angles (30°~120°) by static light scattering (SLS) (CGS-3, ALV, Germany). According to the Guinier Equation (1), the polymer radius of gyration (Rg) of the solution can be calculated through the slope of ln I(q)-q^2^ [36].
(1)$$I(q)=\rho_{0}^{2}v^{2}e^{-\frac{q^{2}R_{g}^{2}}{3}}$$
where q is the scattering vector; I(q) is the scattering intensity; $\rho_{0}$ is the scattering length density; and v is the particle volume.

#### 2.4.4. Fourier-Transform Infrared Spectroscopy (FTIR)

The infrared light interference spectra of the samples, in the range of 400 cm^−1^ to 4000 cm^−1^, were tested by FTIR spectrometer (TENSOR II, Bruker, Karlsruhe, Germany), to characterize the nano-silica modification and the number of hydrogen bonds in the samples. The tested sample was frozen in liquid nitrogen before being ground and pulverized. Then it was mixed with KBr in a certain proportion, and pressed into a transparent film at 10 MPa, for FTIR testing.

#### 2.4.5. Dynamic Mechanical Analysis (DMA)

The entanglement density of the sample can be evaluated through DMA (TA Q850, Newcastle, DE, USA) testing. The PVDF/SiO_2_ composite films (16 mm × 6 mm × 0.4 mm) were heated from −40 °C to 150 °C at a rate of 2 °C/min. A cyclic dynamic strain, with an amplitude of 0.2% and a frequency of 1 Hz, was applied, to test the storage modulus, loss modulus, and tangent value of the samples. The molecular weight between adjacent entanglement points of a material (Me), can be roughly estimated from its storage modulus, according to Equation (2): [37,38]
(2)$$Me=\frac{\rho RT}{G_{N}}$$
where *ρ* is the bulk density of the material at the testing temperature (*T*); *R* is the ideal gas constant; and $G_{N}$ is the plateau portion of the storage modulus. 

#### 2.4.6. Differential Scanning Calorimetry (DSC)

The non-isothermal crystallization process of samples was analyzed by DSC (204 F1, Germany). A 6–8 mg sample was placed in an aluminum crucible, heated from room temperature to 190 °C at a rate of 10 °C/min, held for three minutes to eliminate the thermal history of the sample, and then cooled to 20 °C with the same rate, under a N_2_ atmosphere. The enthalpy change during the cooling process was recorded, and used to calculate the crystallinity of the sample (Xc), through the following Equation (3): [39,40,41]
(3)$$Xc(\%)=\frac{{∆H}_{m}}{{∆H}_{m}^{0}}\;\times \;100\%$$
where ${∆H}_{m}$ and ${∆H}_{m}^{0}$ (104.5 J/g) stand for the crystallization enthalpy of the tested sample and perfect PVDF crystal, respectively. 

#### 2.4.7. Polarized Light Microscopy Test

The sample, placed on a glass slide, was heated to 190 °C on the hot stage of a polarizing microscope (DM2500P, Leica, Weztlar, Germany) for 3 min, and then rapidly cooled to 155 °C at a rate of 50 °C/min. The sample was then maintained at 155 °C, to make it undergo isothermal crystallization, during which the sample was photographed every 30 s to record the crystal growth process.

#### 2.4.8. Mechanical Properties

The mechanical properties of the PVDF/SiO_2_ composite films were measured through a universal test machine (WDL-5kN-II, Guangzhou, China). According to GB/T1040-2018, the test sample was a dumbbell-shaped sample, with a size of 50 mm × 4 mm × 0.5 mm. During the test, the tensile speed was 20 mm/min. The tensile strength, yield strength, elongation at break, and elastic modulus can be determined from the stress–strain curves of the splines.

## 3. Results and Discussion

### 3.1. Diameter of Modified SiO_2_ Nanoparticles

The surfaces of the SiO_2_ nanoparticles were successfully wrapped with a layer of KH570 (confirmed by FTIR, Appendix A), to reduce the surface energy of the nanoparticles and improve the compatibility between the particles and the matrix. We evaluated the modification effect of the particles through DLS and TEM. The DLS results (Figure 1a) show that the three groups of modified SiO_2_ particles have average diameters of 21.19 nm, 31.77 nm, and 56.01 nm, respectively. In addition, Figure 1b–d show TEM images of the three kinds of particles. It can be seen from the TEM images that the diameters of the modified nanoparticles are consistent with those measured by DLS, and the diameter distribution of each group of particles is very uniform (Appendix A). These results indicate that the surface modification of the silica nanoparticles was successful.

### 3.2. Effects of Nano-Silica on the Rheological Behaviors of PVDF Solutions

To more clearly reflect the effect of nanoparticles on the viscosity of the PVDF solution, we configured a concentrated solution, with a mass fraction of 10 wt% of PVDF. The shear viscosity was tested under different shear rates, because the viscosities of polymer solutions are connected to the shear rate [42]. Figure 2a–c show the relationship between the shear viscosity and shear rate of PVDF/SiO_2_ solutions prepared from nanoparticles of different contents and diameters. Overall, the viscosity of the dispersion is almost lower than that of the pure PVDF solution, if the particle diameter is small (20 nm and 30 nm); however, if the dispersed nanoparticles have a larger diameter (50 nm), the viscosities of the dispersion are almost unchanged, or increased compared to the pure PVDF solution, but only a slight decrease occurs when the particle content is 0.8 wt%. Figure 2d demonstrates the variation in the viscosity of the dispersion with particle content after the addition of nanoparticles of various diameters at a low shear rate (0.034 s^−1^). Regardless of the content of the nanoparticles dispersed, the viscosities of the PVDF/30 nm SiO_2_ solutions are always lower than those of the PVDF/20 nm SiO_2_ solutions, whose viscosities are lower than those of the PVDF/50 nm SiO_2_ solutions. In addition, if the diameter of the nanoparticles is around 20 nm or 30 nm, the viscosity of the PVDF solution declines rapidly and then increases gradually with the increase in particle concentration, appearing as a minimum value when the content is 0.5 wt%. However, if the diameter of the nanoparticles is larger (50 nm), the shear viscosity of PVDF remains unchanged with the variation in SiO_2_ particle concentration. The fluctuation range of the shear viscosity is only around 15%. In summary, when the particle content is 0.5 wt% and the diameter is 30 nm, the PVDF/SiO_2_ dispersion can display a minimum viscosity, reduced by up to 76% compared to the pure PVDF solution, at low shear rates.

### 3.3. Dispersion State of Silica (TEM and Rg) 

The TEM images (Figure 3a–c) reflect the dispersion state of the nanoparticles with different diameters in the PVDF matrix at a content of 0.5 wt%. When the nanoparticle diameter is small, the particles show a relatively uniform dispersion state in the matrix. The half distance between the particles dispersed uniformly (h) can be calculated through Formula (4) [20]. When dispersing 30 nm particles into the PVDF matrix, we can calculate the h value as 66.8 nm, which is close to that measured in Figure 3b.
(4)$$\frac{h}{a}={\left(\frac{\Phi_{m}}{\Phi }\right)}^{1/3}-1$$
where a stands for the nanoparticles’ diameter, and $\Phi_{m}$ is the maximum random packing volume fraction, approximately 0.638. When the diameter of the nanoparticles is 50 nm, even with the same addition amount, the nanoparticles still exhibit severe agglomeration in the PVDF matrix, because the thermodynamically stable dispersion state may not be achieved if the diameter is too large. Figure 3d demonstrates the change in *R_g_* before and after adding nanoparticles. In the 10 wt% PVDF solution, the radius of gyration (*R*_*g*0_) of PVDF is 16.93 nm; after dispersing 30 nm nanoparticles, with a content of 0.5 wt%, the radius of gyration (*R_g_*) increases to 17.97 nm (the detailed calculation process of *R_g_* can be found in Appendix A). The increase in *R_g_* is almost negligible, and the actual test results are in good agreement with the theoretical Formula (5) [21,23], so the change in *R_g_* can be attributed to the repulsive volume of the particles.
(5)$$\frac{R_{g}}{R_{g0}}={\left[1+\Phi \right]}^{1/3}$$
where *Φ* is the volume fraction of nanoparticles. The above experimental results, indicate that nanoparticles of a suitable diameter can be well dispersed in the matrix, and enter the entangled network of the polymer, causing the diameter of the polymer to expand slightly. In addition, the radii of 20 nm and 30 nm particles are less than 16.93 nm ($R_{g}$) and can achieve uniform dispersion, while the radius of 50 nm particles is more than $R_{g}$ and will aggregate in the matrix. Therefore, our experimental observation is consistent with Mackay’s work [21].

### 3.4. Effects of Nano-Silica on the Hydrogen Bond of PVDF 

Intermolecular forces are an essential factor affecting the viscosity of polar polymers. For polar PVDF, the fewer the number of intramolecular and intermolecular hydrogen bonds, the lower the viscosity. Therefore, we explored the effect of nanoparticles on the number of hydrogen bonds in the PVDF/SiO_2_ composite, to explain why the addition of nanoparticles led to a reduction in PVDF viscosity. Temperature-dependent FTIR was applied, to study the change in the infrared vibration peak during the heating and cooling process of the sample from 40 °C to 140 °C. Figure 4a–d show the heating and cooling infrared spectra of pure PVDF. The bending vibration peak of –CH_2_– in PVDF should be located at the position of 1376~1380 cm^−1^ [43,44], but it is blue-shifted, due to hydrogen bonding, and hence it is located at 1402 cm^−1^ at 40 °C. However, this peak eventually moves to 1396 cm^−1^ during the heating process, which suggests that the bending vibrational peaks of –CH_2_– tend to be red-shifted as the hydrogen bonds open. At the same time, the stretching vibration peak of –CF_2_–, located at 1183 cm^−1^ [45], gradually blue-shifts to 1192 cm^−1^ with the heating process, which indicates that this peak also undergoes a red-shift in the process of hydrogen bond opening. Figure 4e,f is the IR spectra after adding particles of different diameters. Even if the number of nanoparticles of different diameters is as high as 5 wt%, the peaks at 1402 cm^−1^ and 1183 cm^−1^ do not shift, proving that the addition of nanoparticles does not destroy the hydrogen bonds in the system.

### 3.5. Effects of Nano-Silica on the Entanglement Density of PVDF

Although the molecular chains of PVDF are less flexible, they still have a bit of curling ability, so the degree of entanglement between PVDF molecular chains will also affect the viscosity of PVDF solutions. The storage modulus of PVDF, measured by DMA, can reflect the degree of entanglement between its molecular chains [36,46,47]. The storage modulus can be affected by hydrogen bonding and temperature. There are hydrogen bonds in PVDF, so both the number of hydrogen bonds and the entanglement density will affect the plateau modulus. To eliminate the influence of hydrogen bonds as much as possible, we selected the temperature range where tan δ (δ is the loss angle) was at the lowest point (around 120 °C, Appendix A) as the value of the plateau modulus, according to the previous study [48,49,50], because the number of hydrogen bonds in all samples are almost the same under this condition. In this case, the differences in plateau modulus of different samples come only from the entanglement density. Then, we analyzed the change in storage modulus with temperature for different samples, at the same frequency (Figure 5a). When the content of nanoparticles was 0.5 wt%, the plateau modulus of the three composites declined to a certain extent. In particular, Figure 5b indicates that the storage modulus of the sample is 441.78 MPa, which is only 68.1% of the value of pure PVDF. According to Equation (2), the molecular weight between adjacent entanglement points (Me) of PVDF/30 nm SiO_2_ rises to 13.16 g/mol, but that of pure PVDF is only 8.97 g/mol (Appendix A shows the detailed calculation). The calculated Me is lower than the molecular weight of the monomer unit, because the measured plateau modulus is very high, which is affected by both entanglement density and hydrogen bonds. Therefore, the plateau modulus and Me can only reflect the entanglement density of the samples qualitatively. Macroscopically, the viscosity of the PVDF solution is greatly reduced. Therefore, the fact that the dispersion of nanoparticles with appropriate properties makes PVDF’s viscosity decrease, is because the addition of nanoparticles disentangles the entanglements between molecular chains, rather than reducing the number of hydrogen bonds. These results indicate that dispersing nanoparticles of suitable diameter and content can still reduce the entanglements of the polymer, even in a strongly polar polymer matrix, resulting in a decrease in the viscosity. However, it is worth noting, that the surfaces of the nanoparticles in this experiment are non-polar modified, to avoid the influence of particles on hydrogen bonds, since the viscosity of polar systems is greatly affected by hydrogen bonds. If the particle surface had a polar structure, the polymer would have a strong hydrogen bonding interaction with it, and thus the effect of the particle on the viscosity may be significantly different.

### 3.6. Effects of Nano-Silica on the Crystallization Behaviors of PVDF

Figure 6a–c show the enthalpy change in PVDF/SiO_2_ composites during crystallization. After adding nanoparticles, the initial crystallization temperature of the composite increases from 141.4 °C to 144.9 °C, and the crystallization half-peak width is significantly reduced. The highest crystallinity of PVDF/SiO_2_ composites can reach 39.95%, while that of pure PVDF is 33.5%. The particles, acting as nucleating agents, can induce crystallization, resulting in a shorter crystalline incubation period and faster crystallization speed. At the same time, the crystallization rate and half-peak width of the system change correspondingly with the increase in particle diameter. Combined with the analysis of the crystal morphology, after the isothermal crystallization of different particles (Figure 7a–d), we found that the grain size of PVDF gradually becomes larger and closer to the raw material, as the particle diameter increases, which means that the heterogeneous nucleation effect of nanoparticles on the system gradually weakens when the diameter of particles increases. The smaller the size of the nanoparticles, the larger the specific surface area and the higher the surface energy, allowing for more molecular chains to crystallize on the surface; on the other hand, when dispersing nanoparticles of the same mass (total volume), a greater number of smaller particles can result in more nucleation points. In addition, the uniform distribution of grain size seen in Figure 7, also demonstrates the excellent dispersion of nanoparticles within the composite. Due to a larger number of particles with a diameter of 20 nm, and more nucleation points under the same content, the grain size of PVDF/20 nm SiO_2_ is smaller than other groups. As the particles with a diameter of 50 nm can agglomerate, the grain size of PVDF/50 nm SiO_2_ is not uniform.

### 3.7. Mechanical Properties

The mechanical properties of PVDF were also boosted by dispersing nanoparticles of appropriate diameter and content. Figure 8 shows the mechanical properties of PVDF dispersed with 0.5 wt% SiO_2_ of different diameters. In general, PVDF/SiO_2_ nanocomposites demonstrated better mechanical properties than pure PVDF, which could be ascribed to the Hall–Petch effect [51]. SiO_2_ particles with diameters of 50 nm, can form an agglomeration in the matrix, which can improve the rigidity of PVDF. Therefore, the strength of PVDF is significantly increased, with the yield and tensile strengths reaching as high as 52.1 MPa and 56.4 MPa, respectively. However, the negative effect of agglomeration, is that the toughness of the PVDF becomes poor, and its elongation at break decreases slightly. Particles with a diameter of 20 nm have the effect of grain refinement and thus improve the toughness of PVDF, with a two-fold increase in elongation at break, although its yield strength is slightly reduced. Notably, particles with a diameter of 30 nm can combine both advantages, significantly enhancing the toughness while promoting the strength and rigidity of PVDF. When adding 0.5 wt% 30 nm particles, the elastic modulus and breaking strength of PVDF were as high as 501.6 MPa and 45.9 MPa, respectively, while the elongation at break was 227%. The above experimental results show that dispersing nanoparticles with a content of 0.5 wt% and a diameter of 30 nm, can not only reduce the viscosity of PVDF and improve its processability, but also boost its mechanical properties significantly.

### 3.8. Structure-Rheological Performance Analysis

The dispersion of only a very small number of nano-SiO_2_ can result in a significant reduction in PVDF’s viscosity, without breaking the hydrogen bonds of PVDF, which is noteworthy. Such an anomaly can still be explained by selective adsorption theory, although it has currently only been reported in non-polar polymer systems [15,19]. Nanoparticles with suitable properties could be uniformly dispersed in the polymer matrix, so that they can penetrate the entanglement network of polymer chains, limiting the entanglement density. When dispersing nanoparticles into the solution, polymer chains could be absorbed to the surface of nanoparticles, according to geometric potential theory [52]. For polydisperse polymers, the influence of molecular chains with high molecular weight, on the solution viscosity, is more pronounced. The PVDF investigated in this work has a high polymer dispersity index (PDI), and a small number of particles can have a significant effect on viscosity, so it is inferred that the addition of nanoparticles mainly affects the entanglement degree of polymer chains with high molecular weights. 

The PVDF utilized in this work is a polydisperse system, in which the size of the molecular chains with high molecular weight is denoted as R_g, L_, and the size of other molecular chains is denoted as Rg. Only R_g, L_ could be larger than the particle size when the particle size is 30 nm. Therefore, only PVDF chains with high molecular weight can exhibit good compatibility with nanoparticles. In this case (Figure 9), the nanoparticles will selectively diffuse rapidly into the polymer chains with high molecular weight, and the surface of the nanoparticles will preferentially adsorb PVDF chains of high molecular weight. Compared with 20 nm nanoparticles, the surfaces of 30 nm nanoparticles can adsorb longer molecular chains and exhibit better disentanglement ability. Therefore, the plateau modulus of PVDF/30 nm SiO_2_ should be lower than that of PVDF/20 nm SiO_2_. However, this is not seen in the DMA experiments. This can be explained by the fact that the individual 20 nm nanoparticle is smaller, and there are more of them when adding the same mass, so more molecular chains can be attached to the surface of 20 nm nanoparticles than 30 nm nanoparticles. Thus, the plateau modulus of PVDF/20 nm SiO_2_ and PVDF/30 nm SiO_2_ are very close, meaning they have similar entanglement densities in the PVDF melt (since the DMA tested samples were PVDF sheets). In the state of PVDF melt, the plateau modulus of the sample reflects the disentanglement ability of nanoparticles, and the higher the platform modulus, the weaker the disentanglement ability of the nanoparticles. However, the tested viscosity refers to the viscosity of the polymer solution, in which both the solvent and the nanoparticles have the effect of disentangling the molecular chains. The solvent is more likely to disentangle relatively short chains and the remaining molecular weight for 30 nm particles is less than that for 20 nm, so the solvent disentanglement is more pronounced in PVDF/30 nm samples. As a result, the entanglement density of the PVDF/30 nm SiO_2_ can be greatly reduced, and the macroscopic manifestation of this is the lower viscosity of the PVDF solution. Larger particles (50 nm) are hard to disperse in the matrix, forming large-sized aggregates, which cannot play a role in viscosity reduction. In addition, more PVDF molecular chains can be adsorbed to the surface of the SiO_2_ particles with the increase in particle content, so that the entanglement density between the molecular chains is reduced. However, the dispersion of particles without interaction forces with the matrix, becomes difficult with the further increase in particle content, and they gradually begin to agglomerate. As a result, adding a high content of particles can raise the PVDF viscosity.

## 4. Conclusions

In conclusion, the dependence of PVDF’s rheological properties on nano-silica was investigated by dispersing nanoparticles with different characteristics into 10 wt% PVDF solutions. The viscosity regulation of nanoparticles for non-polar polymers previously observed, is also applicable to PVDF, with strong polarity. When the diameter and content of the dispersed SiO_2_ nanoparticles are 30 nm and 0.5 wt%, respectively, the viscosity of the PVDF/SiO_2_ dispersion can reach a minimum, which is only 24% of that of the pure PVDF solution. Adding nanoparticles into the PVDF can achieve a significant reduction in the viscosity of the PVDF solution, only by reducing the entanglement density of PVDF chains, without breaking its hydrogen bonds, which was proved by DMA and FTIR. Moreover, the TEM and Rg results of the PVDF/SiO_2_ (30 nm, 0.5 wt%), demonstrate the uniform dispersion state of nano-silica in PVDF. Accordingly, the rheological properties of PVDF/SiO_2_ nanocomposites can be explained by the selective adsorption theory. More importantly, compared with pure PVDF, the crystallization starting temperature of the composite is increased by up to 3.5 °C, and the grain size is reduced by up to 9 μm, which indicates nanoparticles can play a role in refining grains and promoting crystallization. PVDF/30 nm SiO_2_ nanocomposites also show the best comprehensive performance, with yield strength and elongation at break as high as 47.5 MPa and 227%, respectively. Therefore, nanoparticles with suitable characteristics can greatly promote the processability of PVDF, while improving its mechanical properties, which is of great significance for practical engineering.

## Data Availability

The data presented in this study are available on request from the corresponding author.

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
