# Peer review of "Rationalizing the Dependence of Poly (Vinylidene Difluoride) (PVDF) Rheological Performance on the Nano-Silica"

_nanomaterials, 2023, doi:10.3390/nano13061096_

Round 1
Reviewer 1 Report
This is my review of the manuscript entitled: "Rationalizing the dependence of poly (vinylidene difluoride) (PVDF) rheological performance on the nano-silica" by Yi Cui, Yang Sui, Peng Wei, Yinan Lv, Chuanbo Cong, Xiaoyu Meng, Hai-Mu Ye, Qiong Zhou (Manuscript ID: nanomaterials-2256377) that explores the influence of nanofillers on the rheological properties of a polar polymer (PVDF).
The authors should read the manuscript anew and correct their grammar. For example p. 3, l. 8 “Characterizations -->Characterization”. Please take the time to correct all of them after a close inspection of your manuscript.
Minor Comments
1) p. 2, l. 3: Ref. [15] is not a paper by Sanjay, please check the reference list to its entirety for potential typos.
2) Sec. 2.3: As usually theories include the volume fractions and not the mass fraction, perhaps include an equation going from one to the other (in essence you would need to provide the densities of each phase).
3) Eq. (2): Please provide Me for your system
4) p. 6, l. 17: You refer to Fig. 3d regarding the value of Rg. Please provide some more explanation as it is not clear, at least to me, how you obtain the Rg values from Fig. 3d
Major Comments
1) p. 1, l. 3: You mentioned of Mackay’s work [21] that experimentally showed that when the particle diameter and spacing are less than the polymer radius of gyration, the system is well dispersed. His work was only an approximate experimental observation, not a theoretical prediction. de Luzuriaga et al. [https://doi.org/10.1002/pat.1029] showed that the spinodal line actually follows an expression of the form Rg~a^n, with n=3/2 (their equation 18), whereas Stephanou [https://doi.org/10.1063/1.4907363] showed that there are actually two branches within which the polymer-NP system is thermodynamically unstable, i.e., immiscible, see his Fig. 2. What do you find from your experimental anaysis?
2) p. 5, l. 15: You refer here to Fig. 2d. As you have not reached the zero-rate limit, I wonder why one would need this comparison. The viscosity reduction presented in the literature, both experimentally but also theoretically, refers to the zero-rate viscosity.
3) p. 8, l. 15-23: You mention here that by adding the nanoparticles, the number of entanglements is reduced. The same is noted by Li et al. [https://doi.org/10.1103/PhysRevLett.109.118001] via simulationa, but beyond a critical volume fraction, they say that the notion of “NP entanglements” comes into play. Do you note something similar at larger volume fractions or have you not studied this? It would be interesting to see what values of volume fractions you have as I guess you are in the dilute regime.
4) p. 9, l. 18-20: “This means that the heterogeneous nucleation effect of nanoparticles on the system gradually weakens when the diameter of particles increases.” I would then guess that nucleation depends on the surface per unit volume ratio: the smaller the particle, the larger this ratio is, allowing for more chains to crystallize on its surface. Could you please comment on this?
Reviewer 2 Report
The manuscript describes the effect of modified SiO2 nanoparticles on some PVDF properties. The experiments are not well described, and the discussion and conclusions are not very convincing. Also, some phrases are quite difficult to understand and the English should be carefully revised.
I list below some observations:
p2, r11: “Meanwhile, polar polymers with strong interaction forces have higher weights in species and market than non-polar polymers, so it is important to explore the law of nanoparticles to the fluidity of strongly polar 13 systems.” – What does it mean that they have higher weights in species and, what is the law of nanoparticles?
p2, r.18: “It has many intramolecular and intermolecular hydrogen bonds, and strong polarity, and its main chain has a certain degree of flexibility [31, 32], which is a typical representative of polar polymers.” How PVDF form intra and intermolecular hydrogen bonds? Which are the atoms involved in these bonds? In the ref. 31 there are hydrogen bonds between PVDF molecules and solvent molecules, but not between polymer molecules. In my opinion some conclusion, like that on page 11, that “dispersing very few SiO2 nanoparticles into the polar PVDF matrix can significantly reduce the solution viscosity of PVDF without breaking its hydrogen bonds.” is confusing.
p2, r33: “PVDF in this study is typically non-polar.” Is it true?
p2, r42: “The solution was first centrifugated…” If there is a supernatant and a precipitate after centrifugation, obviously it is not a solution. The authors also used the term “solution” for their mixtures that are not solutions. A mixture of polymer and silica particles it is not a solution. So please use an appropriate term for the prepared systems.
p2, r. 46: “PVDF/SiO2 composites solution” I think it is wrong to call these composites solutions, here and within the entire manuscript.
p3, r.17: The same for “The nanoparticle ethanol solution” – this is not a solution.
p4, eq.2: What is the use of equation 2? Did the authors calculate the mass between entanglements? If so, please add the obtained values for each sample, not just the increase percent.
p4, r.29: The authors should provide more details about modification of SiO2 nanoparticles surface. What functional groups remain or are added on the particles surface? Are these groups capable of forming hydrogen bonds with de polymer?
p6, r.6: “the dispersion state of the three diameters” I suggest “the dispersion state of the nanoparticles having different diameters.”
p7, r. 10: “Variable temperature infrared…” ????
p11, r.16: “When the particle diameter is 30 nm, only the size of the high molecular weight part (Rg) is larger than the diameter of the nanoparticle (a), so the compatibility of the nanoparticle with the high molecular weight molecular chain will be much better than that of the low molecular weight part (a > Rg).” It is very ambiguous and must be reformulated correctly.
p11, r.21: “Although the polymer chains adsorbed on the surface of 30 nm SiO2 are longer than those on the surface of 20 nm SiO2, the number of 30 nm SiO2 is less than that of 20 nm SiO2 with the same mass.” Please reformulate!
p12, r.5: “The viscosity of the PVDF solution reaches the minimum, only 24% of that of the pure PVDF solution if the diameter and content of nanoparticles are 30 nm and 0.5 wt%, respectively.” ???
Reviewer 3 Report
The paper from Cui et al. thoroughly investigates the influence of silica nanoparticles on the rheological properties of poly(vinylidene difluoride). The paper is quite well written and the conclusions are quite well supported by the experimental data.
However, the manuscript should be revised according to the following points:
- in the Introduction, it could be useful for the Readers to mention the main outcomes concerning the rheological properties of PVDF filled with nanosilica (Ind. Eng. Chem. Res. 2017, 56, 44, 12596–12607) and with “Bud-Branched” Nanotubes (Journal of Macromolecular Science Part B Physics 51:1498–1508, DOI:10.1080/00222348.2011.632742)
- thermogravimetric analyses should be performed in order to better characterize the nanosilica modification
- Figure 8a: please replace "Mpa" with "MPa"
Round 2
Reviewer 2 Report
My comments highlighted in red:
2. p2, r.18: “It has many intramolecular and intermolecular hydrogen bonds, and strong polarity, and its main chain has a certain degree of flexibility [31, 32], which is a typical representative of polar polymers.” How PVDF form intra and intermolecular hydrogen bonds? Which are the atoms involved in these bonds? In the ref. 31 there are hydrogen bonds between PVDF molecules and solvent molecules, but not between polymer molecules.
Response: We greatly appreciate the reviewer’s professional comment. H and F atoms in the same monomer unit can form intramolecular hydrogen bonds, while H and F atoms in different monomer units can form intermolecular hydrogen bonds. The fact that there are strong hydrogen bonds in PVDF can be reflected by the solubility parameter of PVDF. The solubility parameter ( ) consists of three parts: dispersion forces ( ), polar forces ( ), hydrogen bonding forces ( ). The reference [31] indicates (PVDF) = 9.2 , so there should be strong hydrogen bonds in PVDF.
‘It has many intramolecular (formed by H and F atoms in the same monomer unit) and intermolecular hydrogen bonds (formed by H and F atoms in different monomer units, and strong polarity [31], and its main chain has a certain degree of flexibility [32] …’
(Please see line 23-26, page 2)
References:
[31] Bottino, A.; Capannelli, G.; Munari, S.; Turturro, A. Solubility parameters of poly(vinylidene fluoride), J. Polym. Sci. B. Polym. Phys 1988, 26,785-794.
[32] Cui, Z.; Drioli, E.; Lee, Y. M. Recent progress in fluoropolymers for membranes, Prog. Polym. Sci. 2014, 39, 164-198.
Reference [32] does not mention hydrogen bonds within PVDF, while in reference [31] the solubility parameter for PVDF was calculated. In beta-phase PVDF, the intra- and intermolecular interactions consist of attractive dipole-dipole forces. The existence of a hydrogen bonding component of the solubility parameter does not necessarily mean that there are hydrogen bonds inside the polymer, it means that the polymer is capable to form hydrogen bonds when mixed with another component. Compounds like CS2 and CCl4 have a hydrogen bond component of their solubility parameter around 0.6 MPa1/2 and for sure do not form hydrogen bonds. So, the solubility parameters should be used especially when dissolving or mixing of polymers are involved.
Moreover, the modified version:
‘It has many intramolecular (formed by H and F atoms in the same monomer unit) and intermolecular hydrogen bonds (formed by H and F atoms in different monomer units, and strong polarity [31], and its main chain has a certain degree of flexibility [32] …’
is worst then the initial version. Supposing that hydrogen bonds do form, an intramolecular bond would be formed between H and F atoms from the same macromolecule (not monomer unit), while intermolecular bonds would appear between H and F atoms from different macromolecules!!!!
5. p2, r. 46: “PVDF/SiO2 composites solution” I think it is wrong to call these composites solutions, here and within the entire manuscript.
Response: We greatly appreciate the reviewer’s professional comment. Authors of reference [15] wrote that ‘CNTs in solutions can be dispersed by dissolving a polymer
into the suspended solution.’ Therefore, we use “PVDF/SiO2 suspended solution” within the entire manuscript. Please see the revision in the revised manuscript.
I do not understand why reference [15] is used to justify the using of term “solution” in a wrong way. And why the correct terms like “suspension”, “dispersion” are not acceptable for the authors? I do not know what is a suspended solution.
7. p4, eq.2: What is the use of equation 2? Did the authors calculate the mass between entanglements? If so, please add the obtained values for each sample, not just the increase percent.
Response: We really appreciate your professional suggestions and we have provided the Me value in the revised manuscript. The density is around 1.78 g/cm3; R equals 8.314 J/(mol·K); T is 393 K (120℃). Take the pure PVDF as an example to calculate the Me.
The plateau modulus of pure PVDF is 648.72 MPa, and thus:
Me = = =8.97 g/mol
Me (PVDF/20 nm SiO2) = =12.93 g/mol
Me (PVDF/30 nm SiO2) = =13.16 g/mol
Me (PVDF/50 nm SiO2) = =10.99 g/mol
We have provided the Me value in Fig. 5b and the detailed calculation can be seen from the supplementary S6. Please see line 14-17, page 9 in the revised manuscript; highlight in line 16-17, page 4 and line 1-6, page 5 in the supplementary materials.
In my opinion a mass between entanglements that is smaller than the mass of monomer unit has no meaning. How do the authors explain such small values for Me?
Author Response
Dear Reviewer,
Our point-by-point responses are summarized below:
(1) We have cited a new reference that verify the hydrogen bonds in the PVDF and correct our expression in the introduction.
(2) We have explained the small Me value in the section 3.5
(3) We have carefully polished English language of our manuscript and revised the inaccurate expression, such as changing 'suspended solution' into 'dispersion' as suggested by reviewer #2.
Please see the attachment for the 2nd round point-to-point response letter to you.

Reviewer 3 Report
The Authors have addressed the comments arisen by the Reviewer. Now the manuscript can be published in Nanomaterials Journal.
Author Response
Thank you!